# Adaptation of Cyanobacteria to the Endolithic Light Spectrum in Hyper-Arid Deserts

**DOI:** 10.3390/microorganisms10061198

**Published:** 2022-06-11

**Authors:** Bayleigh Murray, Emine Ertekin, Micah Dailey, Nathan T. Soulier, Gaozhong Shen, Donald A. Bryant, Cesar Perez-Fernandez, Jocelyne DiRuggiero

**Affiliations:** 1Department of Biology, The Johns Hopkins University, Baltimore, MD 21218, USA; bmurra17@jhu.edu (B.M.); emertekin@gmail.com (E.E.); micahdailey@gmail.com (M.D.); cperezf2@jhu.edu (C.P.-F.); 2Department of Biochemistry and Molecular Biology, The Pennsylvania State University, State College, PA 16802, USA; riftreiluos@gmail.com (N.T.S.); gxs22@psu.edu (G.S.); dab14@psu.edu (D.A.B.); 3Department of Earth and Planetary Sciences, The Johns Hopkins University, Baltimore, MD 21218, USA

**Keywords:** cyanobacteria, endoliths, desert, far-red-light photoacclimation, metagenome, chlorophylls, photosystems

## Abstract

In hyper-arid deserts, endolithic microbial communities survive in the pore spaces and cracks of rocks, an environment that enhances water retention and filters UV radiation. The rock colonization zone is enriched in far-red light (FRL) and depleted in visible light. This poses a challenge to cyanobacteria, which are the primary producers of endolithic communities. Many species of cyanobacteria are capable of Far-Red-Light Photoacclimation (FaRLiP), a process in which FRL induces the synthesis of specialized chlorophylls and remodeling of the photosynthetic apparatus, providing the ability to grow in FRL. While FaRLiP has been reported in cyanobacteria from various low-light environments, our understanding of light adaptations for endolithic cyanobacteria remains limited. Here, we demonstrated that endolithic *Chroococcidiopsis* isolates from deserts around the world synthesize chlorophyll *f*, an FRL-specialized chlorophyll when FRL is the sole light source. The metagenome-assembled genomes of these isolates encoded chlorophyll *f* synthase and all the genes required to implement the FaRLiP response. We also present evidence of FRL-induced changes to the major light-harvesting complexes of a *Chroococcidiopsis* isolate. These findings indicate that endolithic cyanobacteria from hyper-arid deserts use FRL photoacclimation as an adaptation to the unique light transmission spectrum of their rocky habitat.

## 1. Introduction

In hyper-arid deserts, microorganisms colonize the pore spaces and cracks of translucent rock substrates as a strategy to overcome xeric stress and extreme solar irradiance [1,2]. Substrate properties, such as translucence and pore structure and size, are essential for providing a stable space for colonization that filters high UV radiation, enhances water retention, and allows for photosynthesis [3]. The microbial communities encased in these rocky habitats are called endoliths (within rock) and typically colonize the first few millimeters under the rock surface [4]. Molecular studies of endolithic communities from gypsum, calcite, ignimbrite, sandstone, halite, and granite revealed ecosystems spanning all domains of life, multiple trophic levels, and the presence of diverse viruses [5,6,7,8,9]. Cyanobacteria and algae are the primary producers of endolithic communities and, as such, are essential to these ecosystems [4,6,10,11]. 

Cyanobacteria are oxygenic photosynthesizers that convert photosynthetically visible light (400–700 nm) into chemical energy using complex molecular machinery, including light-harvesting and energy-transducing complexes, pigment biosynthesis, photo-sensory proteins, and associated response regulators [12]. Endolithic cyanobacteria from arid deserts are primarily members of the orders *Chroococcales* (*Chroococcidiopsis* and *Gloeocapsa*), *Nostocales*, and *Oscillatoriales* [2,7,9]. The ability to respond to light properties is essential for phototrophs in the endolithic habitat, where the rock substrate imposes severe physical constraints on light transmission. For instance, light transmission at a wavelength of 665 nm and depth of 2 mm inside calcite, granite, and gypsum rocks varied from 0.01% to 0.1% of the incident light in the Atacama Desert, Chile (2500 µmol photons m^−2^ s^−1^) [4,13]. Values within the same order of magnitude were measured in the interior of halite nodules from the Atacama Desert, with as little as ~0.1 µmol photons m^−2^ s^−1^ deep in the nodule interior [6], and under quartz rocks from the Namib Desert [14]. More remarkable was the differential light transmission across wavelengths for all rocks, resulting in a shift in the transmitted solar spectrum toward far-red light (FRL) [4,6]; this agreed with findings that red-light wavelengths penetrate deeper into rocks than blue-light wavelengths [15].

A variety of light-harvesting strategies to low quantum flux densities and FRL have been described in cyanobacteria, from the synthesis of substituted variants of chlorophyll *a* (Chl *a*) to the spectral tuning of light-harvesting antennae [16,17,18]. In the far-red-light photoacclimation (FaRLiP) response, cyanobacteria exposed to FRL (>700 nm) were found to remodel the core subunits of their photosynthetic apparatus and produce chlorophyll *f* (Chl *f*) and Chl *d*, both red-shifted chlorophylls, allowing the harvesting of photons in the FRL range [19,20,21]. The FaRLiP response is regulated by an FRL-induced signaling cascade composed of RfpA (a knotless phytochrome that senses FRL), RfpB (a DNA-binding response regulator), and RfpC (a signal receiver). These regulators control a gene cluster that includes paralogs for subunits of photosystem I (PSI), photosystem II (PSII), and phycobiliproteins (PBP) [22]. ChlF, a highly divergent paralog of a PSII reaction center protein (PsbA), encodes Chl *f* synthase, a photo-oxidoreductase acting on Chl *a* [17,23]. Phylogenetic analyses suggest that *psb*A4 is likely to have resulted from a duplication of the *psb*A gene [24,25]. Unlike PsbA, Chl *f* synthase cannot bind the Mn_4_Ca_1_O_5_ cluster (preventing water oxidation) but retains a key tyrosine YZ residue and amino acid residues for binding the plastoquinone molecules required for catalytic activity (allowing for the structural change of Chl *a* or chlorophyllide *a*). 

Chl *f*-producing cyanobacteria have been isolated from various low-light environments, including dense microbial mats, lakes, caves, soil, stromatolites, beach rock biofilms, and multiple niches in subtropical forest ecosystems [26,27,28,29]. The wide distribution of Chl *f*-synthesizing cyanobacteria was recently demonstrated using the phycobilisome linker gene *apc*E2 as a marker of FRL-photosynthesis systems [26]. While photosynthetic efficiency is ultimately reduced in the FaRLiP response, it enables cyanobacteria to persist in numerous habitats with low visible light [30]. Despite substantial information about photosynthetic light-harvesting strategies and the associated molecular machinery in many cyanobacteria, our knowledge of light adaptation for endolithic cyanobacteria from hyper-arid deserts remains limited. We used spectroscopy, chromatography, and metagenome annotation to characterize the adaptation of endolithic cyanobacteria we previously isolated from the Atacama and Negev Deserts to the FRL spectrum of their rocky habitat.

## 2. Materials and Methods

*Cyanobacteria cultures and FRL exposure*. Cyanobacterial isolates were obtained from colonized gypsum and calcite samples collected in the Atacama Desert [4] and sandstone samples collected in the Negev desert [10] as described in Murray et al. 2021 [31] (Appendix A). *Microcystis aeruginosa* isolated from Lake Tai, China, was a gift from Feng Chen (IMET, University of Maryland). Cultures were grown in BG-11 liquid medium [32], at RT, under either 24 µmoles photons m^−2^ s^−1^ of visible light (VL) using Philips Daylight Deluxe Linear Fluorescent T12 40-W Light Bulbs and a combination of neutral-density filters (299 1.2ND and 298 0.15ND, Lee Filters, Burbank, CA, USA) or 20 µmol photons m^−2^ s^−1^ of FRL provided by a Flower Initiator 730 nm LED lamp (TotalGrow, Holland, MI, USA). See Appendix A for the spectra of the VL and FRL lamps. 

*Microscopy*. Light microscopy photos were taken from diluted stock cultures under 1000× *g* magnification using an Axioplan II microscope (Zeiss, Jena, Germany) with QImaging QIClick CCD (Teledyne QImaging, British Columbia, Canada) and SensiCam High-Performance cameras (The Cooke Corporation, Romulus, MI, USA). Images were captured and adjusted for clarity with Slidebook 6. 

*Whole-cell analyses*. Whole-cell absorbance spectra from 550 to 800 nm were taken from cells grown in BG-11 under VL and FRL for 30 days using a DU 640 spectrophotometer and associated software (Beckman Coulter, Brea, CA, USA). For low-temperature fluorescence measurements at 77 K, cells resuspended in 60% glycerol were frozen in liquid nitrogen, and spectra were obtained using an SLM Model 8000C spectrofluorometer (On-Line Instruments, Inc., Bogart, GA, USA) [19,22]. Excitation wavelengths of 440 nm and 590 nm were chosen to preferentially excite chlorophylls and phycobilins, respectively [17].

*Pigment extraction and analysis*. Cells of cyanobacterial cultures were harvested by centrifugation, washed once in 50 mM HEPES/NaOH buffer, pH 7.0, and the cell pellets were flash-frozen. Pigments were extracted by resuspending the pellets in a 7:2 (*v*/*v*) ratio of acetone: methanol and vortexing for 2 min. The mixture was centrifuged, and the supernatant was filtered with a 0.2 µm polytetrafluoroethylene membrane before analysis [20]. Room-temperature absorption spectra from extracted pigments were obtained with a Cary 14 UV-Vis-NIR spectrophotometer modified for computer-controlled operation by OLIS Inc. (Bogart, GA, USA). Extracted pigments were also analyzed by reversed-phase high-performance liquid chromatography (HPLC) using an Agilent 1100 HPLC system (Agilent Technologies, Santa Clara, CA, USA), fitted with an analytical Discovery C18 column (25 cm × 4.6 mm) (Supelco, Sigma-Aldrich, St. Louis, MO, USA) [19]. 

*RNA extraction and qRT-PCR*. For the isolation of total RNA from cells of cyanobacterial cultures grown for 48 h under FRL and VL, cells (2.5 mL) were harvested by centrifugation and the pellets flash-frozen. Cell pellets were resuspended in 1 mL of cell lysis buffer (50 mM Tris-HCl, pH 7.5, 25 mM EDTA, 2% sucrose) and 5 µL of SUPERase-in RNase Inhibitor (Invitrogen, Waltham, MA, USA) before cryo-lysis with a SPEX 6870 Freezer Mill (SPEX Sample Prep, Metuchen, NJ, USA) for 5 cycles (1 min grinding at 10 Hz, 1 min cooling). The resulting powder was resuspended in 1 mL of TRIzol LS reagent (Ambion, Inc., Austin, TX, USA) and centrifuged at 4500× *g* for 10 min at 4 °C. Phases were separated by adding 200 µL of chloroform and centrifuged at 4500× *g* for 15 min at 4 °C. RNA in the aqueous phase was precipitated with 2-propanol and centrifugation at 12,000× *g* for 10 min at 4 °C. The RNA pellet was washed with 80% ethanol with centrifugation, resuspended in ddH_2_O, and treated with RNase-free DNase I (New England Biolabs, Ipswich, MA, USA) for 1 h at 37 °C. The Zymo RNA Clean & Concentrator Kit (Zymo Research, Irvine, CA, USA) was used for further purification and concentration of the RNA samples. RNA concentration was measured with the Qubit RNA HS Assay kit (Invitrogen). A mass of 500 ng of total RNA was used to synthesize cDNA with the Invitrogen SuperScript III First-Strand Synthesis System (Invitrogen, Waltham, MA, USA) according to the manufacturer’s recommendations. Primers specific to the *chl*F gene [5′-ATGGTGTCAAAGACAGACA-3′ and 5′-TCATTAGTACTCCAAACCAG-3′] were designed from gene alignments of cyanobacterial isolate metagenomes [31]. Primers for the beta subunit of RNA polymerase gene *rpo*C2 [5′-ATGGTGTCAAAGACAGACA-3′ and 5′-TCATTAGTACTCCAAACCAG-3′] were used for normalization. The PowerUP SYBR Green Master Mix (Applied Biosystems, Waltham, MA, USA) was used to perform Q-RT-PCR with the standard protocol recommended by the manufacturer and a C1000 Touch Thermocycler CFX96 Real-Time System (Bio-Rad Laboratories, Hercules, CA, USA), with the following cycles: 50 °C for 2 min (UDG activation), 95 °C for 2 min, 40 cycles with 95 °C for 15 s, 52 °C (Chl *f* synthase primers) or 58 °C (RpoC2 primers) for 15 s, and 72 °C for 1 min.

*FaRLiP cluster annotation*. To annotate the FaRLip gene cluster, we used the *Chroococcidiopsis thermalis* phycobilisome rod-core linker polypeptide gene (*apc*E2) as a marker [26]. The gene was retrieved from Genbank and blasted against the cyanobacterial metagenome-assembled genomes. The best matches and flanking 25 genes were retrieved to reconstruct the FaRLiP cluster. To annotate the Chl *f* synthase, we performed blastp analysis using a previously characterized Chl *f* synthase protein sequence as a template (UniProtKB/Swiss-Prot: P0DOC9.1). Best matches with >70% amino acid identity were denoted as Chl *f* synthase. A similar analysis was carried out for the RfpABC proteins. The FaRLiP gene cluster was plotted and visualized using the “DNA features viewer” program implemented in python.

## 3. Results and Discussion

The endolithic cyanobacteria used in this work were previously isolated from sandstone rocks from the Negev Desert (S-NGV-2P1), calcite (C-VL-3P3), and gypsum (G-MTQ-3P2) rocks from the Atacama Desert [31]. A significant shift to FRL was reported for the light transmission spectra of these substrates [4]. Taxonomic annotations of most isolates were assigned to the genus *Chroococcidiopsis* based on metagenome sequences [31]. *Microcystis aeruginosa*, isolated from Lake Tai, China, was used as a non-endolithic control. The three endolithic isolates had similar cell morphologies by light microscopy, with large aggregates of cells surrounded by abundant extracellular polymeric substances (EPSs) (Figure 1). Several aggregate morphologies typical to *Chroococcidiopsis* spp. were observed, including first rounds of divisions (Figure 1A), single cells (Figure 1B), and advanced “spore cleavage” that typically follows multiple rounds of division without growth (Figure 1C) [33]. EPSs produced by cyanobacteria in arid environments are essential for retaining moisture and nutrients [34]. These complex heteropolysaccharides prevent water loss by forming a protective shield around the cells and enhance the retention of UV screening compounds [35]. In contrast, *M. aeruginosa*, isolated from a lake, did not form large aggregates and had no visible EPS (Figure 1D); *M. aeruginosa* is known to have diffluent EPSs difficult to visualize under light microscopy. 

### 3.1. Endolithic Cyanobacteria Absorbed FRL Photons 

To determine whether the *Chroococcidiopsis* isolates could absorb FRL photons, we obtained whole-cell absorbance spectra of cultures grown for 30 days in VL and FRL (Figure 2). Spectra for all *Chroococcidiopsis* isolates and *M. aeruginosa* grown in both light conditions showed an absorbance peak at 680 nm indicative of Chl *a*. A shoulder at 710 nm, indicative of the presence of Chl *f* [17], was only found in the spectra of endoliths grown in FRL. Chl *f* is a specialized chlorophyll synthesized from Chl *a* during the FaRLiP response, and it differs from Chl *a* by the presence of a formyl group at the C2 position [36]. In all *Chroococcidiopsis* spp. cultures, there was greater overall absorbance between 600 nm and 700 nm in VL-grown cells when compared to FRL-grown cells, possibly in response to the higher photon flux at those wavelengths in VL conditions. 

### 3.2. Pigments from FRL-Grown Chroococcidiopsis Contained Chl f and Chl d

The G-MTQ-3P2 *Chroococcidiopsis* spp. isolate was selected for further pigment analysis. To validate the presence of the FRL-absorbing Chls (chlorophylls) in those cultures, we conducted analyses on methanol-extracted pigments from cultures grown for 30 days in VL and FRL. Similar to whole cells, the absorbance spectrum of pigments from cultures grown in FRL displayed the characteristic Chl *f* shoulder at ~705 nm [19,27], which was not found in the pigments from VL-grown cultures (Figure 3A). In addition to the shoulder at 705 nm, we found an absorbance peak at 770 nm in pigments from the FRL-grown culture, which is characteristic of bacteriochlorophyll *a* (Bchl *a*) [37]. Reversed-phase HPLC of pigments from VL and FRL-grown cultures showed the presence of Chl *a*, eluting at ~47 min under both light conditions. Chl *f*, eluting at ~43 min, was only found in Chls extracted from FRL-grown cultures (Figure 3B). BChl *a* and a small amount of Chl *d*, another “red-shifted” chlorophyll, were also detected in the pigments from cells grown in FRL. FRL activates the synthesis of Chl *d* in addition to Chl *f* [38], supporting our finding of a small amount of Chl *d* in FRL-grown cultures of the *Chroococcidiopsis* G-MTQ-3P2 isolate (Figure 3B). 

FRL-activated photosystems were reported to have 7 Chl *f* out of 90 Chls in PSI and 4 Chl *f* and 1 Chl *d* out of 35 Chls in PSII [39,40,41,42,43]. We used values from the absorbance spectra of Figure 3A to calculate the ratio of Chl *f*: Chl *a* in G-MTQ-3P2 cells grown in FRL. The fraction of Chl *f* absorption in the 663 nm peak in FRL was removed by calculating the fraction of Chl *f* absorption in the 663 nm peak versus the 705 nm peak, using a Chl *f* 663/705 ratio of 0.150 calculated from Airs et al. [44], and by subtracting that fraction from the 663 nm absorption value in FRL. Extinction coefficients for Chl *a* (78.8 × 103 L mol^−1^ cm^−1^) at 663 nm and Chl *f* (77.97 × 103 L mol^−1^ cm^−1^) at 705 nm were used to calculate their respective amounts. We calculated a ratio of Chl *f*: Chl *a* of 1:17.6 for the *Chroococcidiopsis* G-MTQ-3P2 isolate, which was slightly higher than the value reported for *Chlorogloeopsis. fritschii* [44] and in line with the reported number of Chl *f* in FRL-acclimated PSII [39,40,41,42]. While the assembly and function of PSII in FRL require Chl *d*, the enzyme(s) performing the synthesis of Chl *d* remains unknown. Recent work with *Synechococcus* sp. PCC 7335 *apc* mutant strains suggested that cysteine-rich FRL allophycocyanin subunits might play a role in the synthesis or stabilization of the single Chl *d* molecule required for the assembly of PSII under FRL [38,43]. The findings that *Chroococcidiopsis* sp. isolate G-MTQ-3P2 synthesized Chl *f* under FRL, together with a small amount of Chl *d*, strongly suggested that endolithic cyanobacteria might use the FaRLiP response to adapt to the red-shifted light of their rock environment.

The BChl *a* found in FRL-pigments is indicative of the presence of another bacterium in the culture of the *Chroococcidiopsis* G-MTQ-3P2 isolate. The absence of other BChls in the HPLC pigment analysis (Figure 3B), and the peak at 770 nm in the pigment absorption spectrum (Figure 3A), suggested that the photoheterotrophic bacterium in the G-MTQ-3P2 culture might belong to the purple bacteria [37]. These are anoxygenic photoheterotrophs from the phylum *Proteobacteria* with versatile metabolic capabilities [45]. The metagenome-assembled genome (MAG) from the G-MTQ-3P2 metagenome was assigned to *Chroococcidiopsis* through taxonomic annotation [31]. Additional contigs were annotated with BLAST and assigned to heterotrophic bacteria of the *Actinobacteria* (43.3%), *Proteobacteria* (<1%), and *Deinococcus* (<1%) phyla. While no purple bacteria were specifically annotated in the G-MTQ-3P2 metagenome, it is important to note that the metagenome was sequenced from DNA extracted from cultures grown in VL, potentially explaining the low relative abundance of putative Bchl *a*-producing proteobacteria. Under FRL, purple bacteria can grow photoautotrophically at low oxygen levels [45], providing them with a competitive advantage. Because the cultures were grown without agitation, the cells likely settled to the bottom of the culture flasks forming a biofilm where low oxygen concentrations allowed them to grow.

### 3.3. PSI, PSII, and PBS Have Different Spectral Properties after FRL Exposure 

To determine whether endolithic cyanobacteria remodeled their photosynthetic machinery under FRL, we used 77 K low-temperature fluorescence emission spectroscopy on G-MTQ-3P2 cultures grown in VL and FRL. The fluorescence emission spectrum with 440 nm excitation, optimized for Chl excitation, for VL-grown cultures showed emission peaks at 684 and 723 nm, indicating the association of Chl *a* with PSII and PSI, respectively (Figure 4A) [20]. In the FRL-spectrum, the 684 nm peak was larger, and there was no 723 nm peak (Figure 4A). The increased signal of the Chl *a*-PSII complex in FRL was also observed in FRL-grown *Synechococcus* sp. PCC 7335 but not *Chlorogloeopsis* sp. PCC 9212 or *Leptolyngbya* sp. JSC-1 [21]. The FRL-spectrum of G-MTQ-3P2 contained peaks at 738 nm and 749 nm contributed from the Chl *f* and *d*-containing FRL-PS II and Chl *f*-containing FRL-PS I complexes, respectively [21]. A similar peak at 749 nm was also reported with FaRLiP acclimation of two other cyanobacteria, *Chroococcidiopsis thermalis* PCC 7203 and *Fischerella thermalis* PCC 7521 [19]. The 718 nm peak observed in the G-MTQ-3P2 FRL-spectrum might be related to the fluorescence emission from FRL-AP cores; however, because the 440 nm wavelength mostly excites Chls, there might also be fluorescence emission from some remaining VL-PSI. 

Low-temperature fluorescence emission spectroscopy with 590 nm excitation preferentially excites PBP. The energy transfer from these proteins to the photosystems provides another way to investigate changes to the light-harvesting machinery under FRL. At 590 nm excitation in VL-grown cells, we observed peaks at 660 and 682 nm, corresponding to allophycocyanin and ApcD and ApcE terminal emitters of the phycobilisome, respectively, and an additional peak at 721 nm (Figure 4B). In contrast, in FRL, peaks of key phycobiliproteins, including allophycocyanin, and the PBS terminal emitter were red-shifted by a few nanometers. With 590 nm excitation, a 717 nm peak thought to be indicative of energy transfer from allophycocyanin to the Chl *f*-PSII complex [21] was weakly present in the FRL-grown cultures of *Chroococcidiopsis* G-MTQ-3P2 (Figure 4B). However, the shoulder (VL) and the small peak at 648 nm (FRL) (Figure 4B) were indicative of phycocyanin; the fact that there was little phycocyanin emission indicates that that was not a large pool of unassembled phycocyanin in the cytoplasm and that most of the phycocyanin was coupled for energy transfer to PSI and PSII. In the phycobilisome of FRL-grown *Leptolyngbya* sp. JSC-1 [46], pentacylindrical cores were replaced by bicylindrical cores as indicated by a peak at 730 nm. While we did not find a peak at 730 nm with G-MTQ-3P2, we found a peak at 750 nm that was indicative of the presence of long-wavelength chlorophylls in the PSI core in FRL-grown *C. thermalis* [41]. These differences between strains indicated that they most likely used different strategies for remodeling their light-harvesting complexes in FRL.

### 3.4. FaRLiP Gene Clusters Identified in Endolithic Chroococcidiopsis Metagenomes

Using multiple sequence alignments of amino acids, we identified the Chl *f* synthase gene (*chl*F) in metagenome-assembled genomes (MAGs) from all endolithic *Chroococcidiopsis* spp. isolates (Appendix A). This gene is essential for the FaRLiP response because it encodes the photo-oxidoreductase that synthesizes Chl *f* from Chl *a* [17,23]. An amino acid alignment of ChlF from *C. fritschii* PCC 9212 with all PsbA proteins annotated in the G-MTQ-3P2 metagenome (Appendix A) revealed that the endolithic ChlF sequences lack the three key amino acid residues required for binding the Mn_4_Ca_1_O_5_ cluster, indicating that these proteins were not functional D1 core subunits of PSII [17]. The endolithic *Chroococcidiopsis* ChlF sequences were identical and shared 72% amino acid identity with ChlF from *Chlorogloeopsis* sp. PCC 9212 and 56% identity with ChlF of *Synechococcus* sp. PCC 7335. 

Functional annotation and BLAST analysis revealed the presence of FaRLiP-associated genes in all the *Chroococcidiopsis* spp. MAGs with an identical organization (Figure 5, Appendix A). These included paralogs of PSII, PSI, PBP, and the FaRLiP regulatory elements RfpB, RfpA, and RfpC [22]. Several hypothetical proteins were also found in the same genomic neighborhood. ApcE2, a gene encoding an FRL-associated phycobilisome linker and thought to be a marker of the FaRLiP response [26], was also found in the *Chroococcidiopsis* MAGs. The arrangement of the FaRLiP cluster from endolithic *Chroococcidiopsis* was nearly identical to that of *C. thermalis* PCC 7203. Both contain four paralogs of PSI genes, while six are found in many other species, including *Synechococcus* sp. PCC 7335, *Leptolyngbya* sp. JSC-1, and *Chlorogleopsis* sp. PCC 9212 [19]. As in all of these species, except *Leptolyngbya*, PSII paralogs flank the PBS genes and are located downstream of the regulatory *rfp*ABC genes (Figure 5). 

### 3.5. The chlF Gene Was Only Expressed in FRL-Grown Cultures

We used qRT-PCR to test whether the endolithic *chl*F gene found in the genome of *Chroococcidiopsis* G-MTQ-3P2 was exclusively expressed in FRL-grown cells. We used Chl *f* synthase (*chl*F) gene-specific primers, designed from the multiple alignments of endolithic *chl*F genes, and primers for RNA polymerase protein *rpoC2*, as a housekeeping gene. Templates include RNA isolated from three biological replicates of *Chroococcidiopsis* G-MTQ-3P2 grown in both VL and FRL for 48 h. We found that the *chl*F gene was expressed exclusively in FRL-grown cells, whereas the *rpoC2* gene was expressed in both light conditions (Figure 6), confirming the function of ChlF in the FaRLiP response of endolithic *Chroococcidiopsis*. Ct values for *chl*F were comparable to values for *rpoC2*, indicating nonnegligible levels of expression for *chl*F in FRL.

## 4. Conclusions

Our characterization of several endolithic cyanobacteria isolated from the Atacama and Negev deserts revealed that they all encoded genes from the FaRLiP cluster, expressed Chl *f* synthase under FRL, and therefore used far-red-light photoacclimation to perform oxygenic photosynthesis. Hyperspectral imaging previously demonstrated that the distribution of Chl *f* was related to the position of the cyanobacterial colonies in beach rock biofilms [29]. This work further provides evidence for the stratification of Chls and photosynthetic activity throughout lithic substrates [11]. Surprisingly, FaRLiP acclimation was not detected in hypolithic cyanobacteria from the Namib Desert [47]. Hypolithons are microbial communities colonizing the underside of quartz rocks and they are ubiquitous in arid deserts around the world. Instead, the presence of several orange carotenoid-like proteins (OCPs) in hypolithic cyanobacteria was suggested as a potential photoprotection against sudden changes in light influxes and as a protection against desiccation stress [47]. Hyper-arid environments impose multiple challenges to microbial life and endolithic cyanobacteria have evolved numerous survival strategies. While the rock substrate provides UV screening and enhanced water retention, it also filters photosynthetically active radiation required for primary production in the endolithic habitat. Despite some decrease in photosynthetic efficiency [30], we showed that FRL-acclimated photosynthetic proteins and pigments allowed endolithic *Chroococcidiopsis* spp. to optimize light-dependent energy production in their rocky habitat.

## Figures and Tables

**Figure 1 microorganisms-10-01198-f001:**
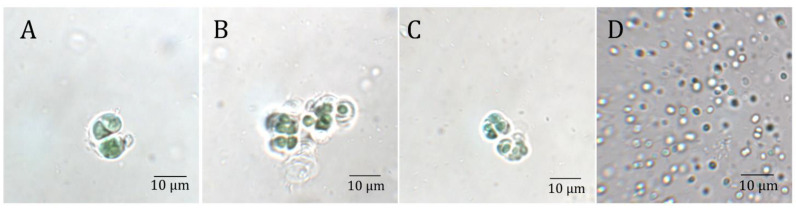
Light microscopy images of cyanobacterial isolates from several substrates and locations: (**A**) Sandstone from the Negev desert (S-NGV-2P1), (**B**) Calcite (C-VL-3P3) and (**C**) Gypsum (G-MTQ-3P2) from the Atacama Desert, and (**D**) *M. aeruginosa* isolated from Lake Tai, China.

**Figure 2 microorganisms-10-01198-f002:**
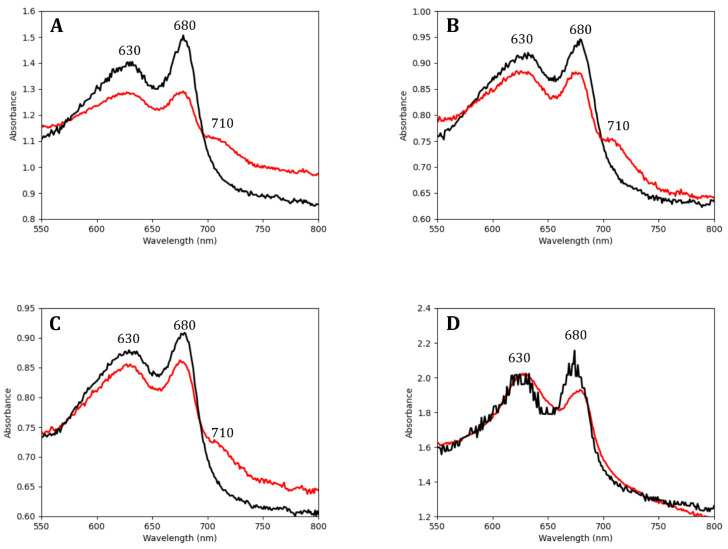
Comparison of absorbance spectra from whole-cell cultures grown under visible light (black line) and far-red light (red line) for *Chroococcidiopsis* sp. isolates from (**A**) calcite (C-VL-3P3), (**B**) gypsum (G-MTQ-3P2), (**C**) sandstone (S-NGV-2P2), and (**D**) *M. aeruginosa* from Lake Tai. Absorption peaks for PBP (at 630 nm), Chl *a* (at 680 nm), and Chl *f* (at 710 nm) are labeled. Cultures were grown under VL or FRL for 30 days before the experiment. Peaks were normalized to OD_575_. Each spectrum was the average of three measurements.

**Figure 3 microorganisms-10-01198-f003:**
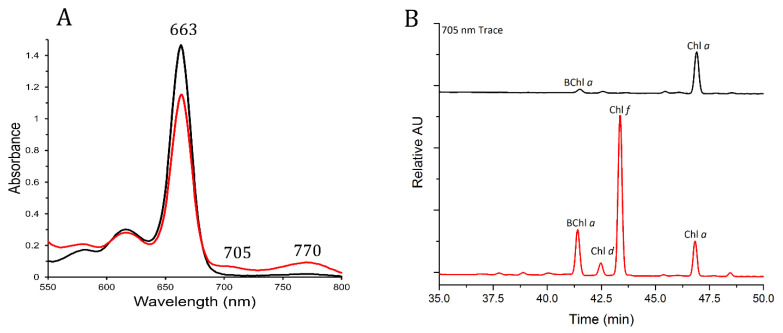
Absorption spectroscopy and HPLC analyses of pigments extracted from *Chroococcidiopsis* G-MTQ_3P2 cultures grown in VL (black) and FRL (red). Pigments were extracted with a mixture of acetone and methanol. (**A**) Absorption spectra of total pigments from 550 to 800 nm. Labels indicate the maxima for Chl *a*, Chl *d*, Chl *f*, and BChl *a*. (**B**) Reversed-phase HPLC elution profiles at 705 nm; BChl *a* eluted at about 41 min, Chl *d* at about 42.5 min, Chl *f* at about 43.5 min, and Chl *a* at about 47 min under these conditions. Each spectrum was the average of three measurements.

**Figure 4 microorganisms-10-01198-f004:**
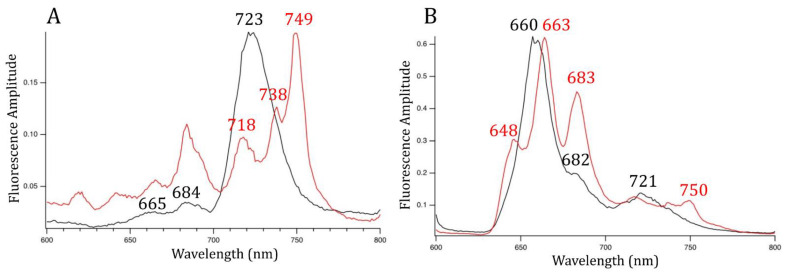
77 K fluorescence emission spectra for *Chroococcidiopsis* G-MTQ-3P2 cultures grown in VL (black line) and FRL (red line). (**A**) Excitation wavelength at 440 nm to excite Chls preferentially. (**B**) Excitation wavelength at 590 nm to excite PBPs preferentially. Cells were resuspended in 60% glycerol at OD_780_ of 0.7. Each spectrum was the average of three measurements.

**Figure 5 microorganisms-10-01198-f005:**
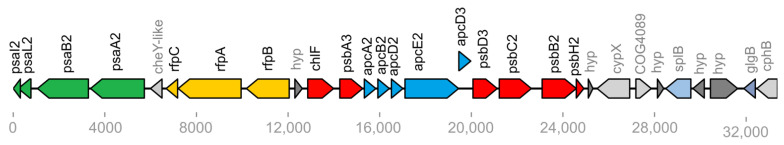
Gene map of the FaRLiP gene cluster found in the MAG of *Chroococcidiopsis* G-MTQ-3P2. Paralogs of PSII are in red, paralogs of PSII are in green, paralogs of phycobiliproteins are in blue, and regulatory genes are in yellow. Hypothetical genes are in dark grey, and genes involving other processes are in light blue and light grey.

**Figure 6 microorganisms-10-01198-f006:**
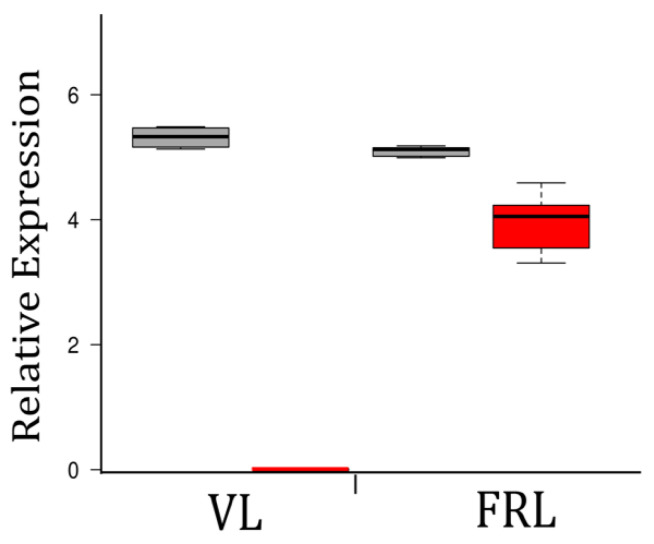
Chl *f* synthase expression under VL and FRL in *Chroococcidiopsis* G-MTQ-3P2 cultures. Box and whisker plot of relative expression values (1/Ct × 100) from qRT-PCR for the *rpoC*2 gene (grey) and the Chl *f* synthase, *chl*F gene (red). Three biological replicates and three technical replicates were used for each condition. The lines in the center of each box represent the median value. The top and bottom of each box represent the 25th and 75th percentiles, respectively. Whiskers extend to 1.5 times the interquartile range between these limits. N = 9 sample points.

## Data Availability

The metagenome-assembled genomes (MAGs) and functional annotation are available from the JGI Genome Portal under the IMG taxon #3300037877 for G-MTQ-3P2, 3300039404 for C-VL-3P3, and 3300039401 for S-NGV-2P1 {Murray et al., 2021 [31]} (Appendix A).

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
