# Peer review of "Adaptation of Cyanobacteria to the Endolithic Light Spectrum in Hyper-Arid Deserts"

_microorganisms, 2022, doi:10.3390/microorganisms10061198_

Round 1
Reviewer 1 Report
The article is devoted to the study of algae capable of photosynthesis under conditions of far red light with the participation of chlorophyll f.
The authors did a great job, got interesting data. The applied methods are adequate.
There are wishes for the design of the gene expression pattern, and express it in relative expression units or take the logarithm of the data if the differences are large.
Why was the expression data not expressed in delta delta Cq?
Do you need to explain in more detail why you decided not to give expression to the other two strains of algae?
Author Response
We appreciate the positive feedback on our study.
There are wishes for the design of the gene expression pattern, and express it in relative expression units or take the logarithm of the data if the differences are large.
Why was the expression data not expressed in delta delta Cq?
We could not calculate the delta delta Ct values because we did not get any product with the Ch f synthase primers with RNA from cells grown in VL.
Do you need to explain in more detail why you decided not to give expression to the other two strains of algae?
We had limited space in our FRL incubator to grow all the strains in triplicate cultures. Because the FaRLiP operon was identical across all isolates, we decided against doing multiple rounds of culturing.
Reviewer 2 Report
Review comments on “Adaptation of cyanobacteria to the endolithic light spectrum in hyper-arid deserts”
General comment:
Certainly, the study has brought new knowledge about Chl f biosynthesis and its adaptation role in endolithic environments where the sole light source is far-red light. The manuscript is well written with appropriate experiments with supporting data and discussion, however, there are a few minor corrections, which need to be fixed.
Specific comments:
Lines 20 and 24: Correct the typographical error for “Chroococidiopsis”
Line137: Correct the error “resuspended in 1 mL μl of TRIzol LS reagent….”
Line209: Delete “spp.” Because this part of the study relates to a single “isolate” and not multiple “sp.”
Paragraph292-307: What about “phycocyanin” which is one of the main core components of cyanobacterial phycobilisomes? Did Fluorescent emission spectra suggest any PC presence in any of the tested light conditions?
Figure1 Legend: Check for the error in the sentence. Also if available, provide high-quality images with a distinct morphology.
Figure3: A: Label the peaks with appropriate “nm”. B: label “Chl d” in the elution profile at
Figure6: Obviously chlF gene expression is significantly induced in FRL, but not sure, what would be the explanation for the slightly different expression of the housekeeping rpoc2 gene?
TableS2: Column 3 for gene names should be italicized appropriately
Figure S1: label the spectra for VL and FRL lamps with arrows
Author Response
We appreciate the positive feedback from this reviewer.
Lines 20 and 24: Correct the typographical error for “Chroococidiopsis”
Done
Line137: Correct the error “resuspended in 1 mL μl of TRIzol LS reagent….”
Done
Line209: Delete “spp.” Because this part of the study relates to a single “isolate” and not multiple “sp.”
Done
Paragraph292-307: What about “phycocyanin” which is one of the main core components of cyanobacterial phycobilisomes? Did Fluorescent emission spectra suggest any PC presence in any of the tested light conditions?
There was some; here is what we added to the text to that regard: “However, the shoulder (VL) and the small peak at 648 nm (FRL) (Fig. 4B) were indicative of phycocyanin; the fact that there was little phycocyanin emission indicates that was not a large pool of unassembled phycocyanin in the cytoplasm and that most of the phycocyanin was coupled for energy transfer to PSI and PSII”
Figure1 Legend: Check for the error in the sentence. Also if available, provide high-quality images with a distinct morphology.
Done. High-quality images were provided with the revised version of the MS
Figure3: A: Label the peaks with appropriate “nm”. B: label “Chl d” in the elution profile at
We added 663, 705, and 770 to panel A to indicate the maxima for Chl a, Chl f, and BChl a. We do not feel comfortable with labeling the Chl d peak at the sensitivity level displayed; it is likely that we detected Chl d but will need a much higher sensitivity level to confirm. We also remove mention of Chl d peak from the legend.
Figure6: Obviously chlF gene expression is significantly induced in FRL, but not sure, what would be the explanation for the slightly different expression of the housekeeping rpoc2 gene?
We did not detect any signal with the Ch f synthase primers and RNA from cells grown in VL; as such we could not calculate the delta delta Ct values. The small difference in rpoc2 levels between VL and FRL is most likely because of experimental inaccuracies (i.e. RNA concentrations, pipetting,..).
TableS2: Column 3 for gene names should be italicized appropriately
Done
Figure S1: label the spectra for VL and FRL lamps with arrows
I am not sure why there there are multiple lines for one light condition and only one for FRL. This might be addressed by labeling the figure or by calling out the line colors. The figure would be improved by increasing the width of the lines. It is otherwise difficult to see the lines.
Done; we called out line numbers and absorption maxima.
Reviewer 3 Report
Well written and well planned work. The conclusions of the paper are fully justified. I don't have any significant comments.
A few small notes:
1) Lines 176-185: While speaking about EPS, please, be more accurate. Chroococcidiopsis has EPS in the form of dense capsules, so they are clearly visible under light microscopy. But Microcystis has a different type of EPS - free or diffluent EPS which is not clearly seen under light microsopy. Thus, I believe that it would be more correct to write about different types of EPS in these microorganisms, and not about the presence of a clearly visible EPS in Chroococcidiopsis and the absence of a visible EPS in Microcystis.
2) Line 203: There is a mistake in the strain number: Gypsum (G-MTQ-2P3). Please, correct.
3) Lines 299-301: The autors write "With 590 nm excitation, a 717 nm peak thought to be indicative of energy transfer from allophycocyanin to the Chl f-PSII complex [21] was not found in the FRL-grown cultures of Chroococcidiopsis G-MTQ-3P2 (Fig. 4B)". But I see a small red line peak slightly shifted to the left from 721 nm black line peak in the Figure 4B. Does this peak correspond to 717 nm or not? Please explain.
Author Response
We appreciate the positive feedback on our study.
1) Lines 176-185: While speaking about EPS, please, be more accurate. Chroococcidiopsis has EPS in the form of dense capsules, so they are clearly visible under light microscopy. But Microcystis has a different type of EPS - free or diffluent EPS which is not clearly seen under light microsopy. Thus, I believe that it would be more correct to write about different types of EPS in these microorganisms, and not about the presence of a clearly visible EPS in Chroococcidiopsis and the absence of a visible EPS in Microcystis.
Thank you for pointing this out; we added the following to the MS: “M. aeruginosa is known to have diffluent EPSs difficult to visualize under light microscopy.”
2) Line 203: There is a mistake in the strain number: Gypsum (G-MTQ-2P3). Please, correct.
Done.
3) Lines 299-301: The autors write "With 590 nm excitation, a 717 nm peak thought to be indicative of energy transfer from allophycocyanin to the Chl f-PSII complex [21] was not found in the FRL-grown cultures of Chroococcidiopsis G-MTQ-3P2 (Fig. 4B)". But I see a small red line peak slightly shifted to the left from 721 nm black line peak in the Figure 4B. Does this peak correspond to 717 nm or not? Please explain.
The reviewer is right and we changed the MS to” With 590 nm excitation, a 717 nm peak thought to be indicative of energy transfer from allophycocyanin to the Chl f-PSII complex [21] was weakly present in the FRL-grown cultures of Chroococcidiopsis G-MTQ-3P2 (Fig. 4B).”
Reviewer 4 Report
A brief summary
Light adaptation to far red light (FRL) in the endocytic cyanobacterium Chroococidiopsis spp. was investigated. When cultivated on FRL, Chroococidiopsis isolates synthesized a significant amount of long-wavelength chlorophylls f and d. Сhanging the composition of photosynthetic proteins and pigments under FRL growth allows Chroococcidiopsis to optimize light-dependent energy production in their rocky habitat.
Broad comments
The results of this publication are of undoubted interest to specialists in photosynthesis, biologists studying extremophilic organisms, and astrobiologists.
Specific comments
I was unable to open the _Murray_Cyano_suppMaterial_v2 - try to simplify the procedure for opening it.
Line 205: «Peaks were normalized to OD700». These two curves have the same values of OD700 only in Figure 2 B. I recommend to correct Figures 2 A, C and D or change the caption.
Author Response
I was unable to open the _Murray_Cyano_suppMaterial_v2 - try to simplify the procedure for opening it.
We are sorry the reviewer was unable to open the file for supplementary material. As requested by the journal, we uploaded a word document similar to that of the MS.
Line 205: «Peaks were normalized to OD700». These two curves have the same values of OD700 only in Figure 2 B. I recommend to correct Figures 2 A, C and D or change the caption.
Thank you for pointing this out, the peaks for each channel were normalized to OD575 and not OD700. We made the appropriate changes to the MS.